# Assessment of Tree Diameter Estimation Methods from Mobile Laser Scanning in a Historic Garden



Enrique Pérez-Martín [1,*] , Serafín López-Cuervo Medina [2] , Tomás Herrero-Tejedor [1] , Miguel Angel Pérez-Souza [1] , Julian Aguirre de Mata [2] and Alejandra Ezquerra-Canalejo [3]

1    Departamento de Ingeniería Agroforestal, Universidad Politécnica de Madrid, 28040 Madrid, Spain; tomas.herrero.tejedor@upm.es (T.H.-T.); miguelangel.perez.souza@upm.es (M.A.P.-S.)

2    Departamento de Ingeniería Topográfica y Cartográfica, Universidad Politécnica de Madrid, 28031 Madrid, Spain; s.lopezc@upm.es (S.L.-C.M.); julian.aguirre@upm.es (J.A.d.M.)

3    Departamento de Ingeniería y Gestión Forestal y Ambiental, Universidad Politécnica de Madrid, 28040 Madrid, Spain; alejandra.ezquerra@upm.es

*    Correspondence: enrique.perez@upm.es; Tel.: +34-910-670-975

**Abstract:** Geo-referenced 3D models are currently in demand as an initial knowledge base for cultural heritage projects and forest inventories. The mobile laser scanning (MLS) used for geo-referenced 3D models offers ever greater efficiency in the acquisition of 3D data and their subsequent application in the fields of forestry. In this study, we have analysed the performance of an MLS with simultaneous localisation and mapping technology (SLAM) for compiling a tree inventory in a historic garden, and we assessed the accuracy of the estimates of diameter at breast height (DBH, a height of 1.30 m) calculated from three fitting algorithms: RANSAC, Monte Carlo, and Optimal Circle. The reference sample used was 378 trees from the Island Garden, a historic garden and UNESCO World Heritage site in Aranjuez, Spain. The time taken to acquire the data by MLS was 27 min 37 s, in an area of 2.38 ha. The best results were obtained with the Monte Carlo fitting algorithm, which was able to estimate the DBH of 77% of the 378 trees in the study, with a root mean squared error (RMSE) of 5.31 cm and a bias of 1.23 cm. The proposed methodology enabled a supervised detection of the trees and automatically estimated the DBH of most trees in the study, making this a useful tool for the management and conservation of a historic garden.

**Keywords:** tree inventory; point cloud; terrestrial laser scanning; SLAM; TLS; diameter at breast height (DBH); historical garden

## 1. Introduction

Compiling inventories of elements and cataloguing heritage spaces are all necessary operations for preserving any cultural landscape [1].

Some cultural spaces that have been designated as historic gardens have singular characteristics that hinder the use of new sensors to compile tree inventories. Landscape elements, such as the morphology of paths and hedges, signage, and ornamental fountains and statues, cause occlusion when collecting data by laser scanner [2], so an essential phase of this study was to develop an adequate methodology, including the optimum planning of the data acquisition process.

Geo-referenced 3D models can be used as a baseline data base for cultural heritage projects and forest inventories [3,4]. The latest instruments and geomatic techniques offer new developments in sensorisation and an easy workflow for 3D metric documentation. The visualisation of a highly detailed 3D model brings a new perspective to topographic mapping, analysis and interpretation, geometric and assessment studies, multi-temporal measurements, virtual reconstructions, and immersive reality. In recent years, theuse of terrestrial laser scanning (TLS) has raised the possibility of obtaining 3D models [5] from point clouds, which are now used in a variety of areas [6], including forest inventories [7–10] and forest

structure [11–14] and classification [12]. TLS scans are made individually from a fixed point or by integrating several scans with common points. The settings and scanning modes of the TLS affect the efficiency with which the trees are detected [15,16]; the denser the point cloud, the more effective the detection [13]. Based on forest inventories compiled using TLS, Koren et al. [17] report that research into forest modelling, volume, biomass measuring, and temporal studies are opening up new opportunities in the field of silviculture. TLS combined with structure from motion (SfM) terrestrial photogrammetry techniques have been used to carry out forest inventories [18]. Recent studies [19] have assessed the use of smartphones to compile forest inventories through relative positioning without global navigation satellite system (GNSS) signals. Studies comparing TLS and MLS based on SLAM for use in forest inventories on various terrains revealed that MLS detected 96% of the trees, whereas TLS detected 78.5% [20], and took less time to collect the data. The use of TLS requires multiple scanning bases to ensure the effective detection of the trees, and the most significant problems are the effects of shade or concealment by trees [14,15] due to the range of the scanner [21]. Some studies on the quality of the point cloud obtained by MLS report a problem in the model due to noise [14] or errors in fitting the geometric shapes [22].

Using MLS to generate 3D point clouds offers a means of obtaining precise and reliable models thanks to the speed and ease with which data can be collected in the field. Its application has recently been assessed when inventorying trees on urban land [23,24], with the added difficulty that some specimens were occluded in the trajectory. MLS significantly reduces the time and cost of collecting data in the field compared to other geomatic techniques, and minimises the problem of detection errors due to shadows or concealment. High-density point clouds can characterise trunks in tree stands, while other studies address the calculation of diameter at breast height [21,25–28], tree shape and size [29], the characterisation of the root system [30], scanning parameters [31], etc.

The diameter of a tree at a height of 1.30 m (DBH) from the ground is the most widely used characteristic when compiling forest inventories and for estimating the volume of timber and biomass in trees [32], and is traditionally measured with a measuring tape or a tree calliper. The availability of sensors such as TLS or MLS has led to the search for an effective and rapid methodology to collect this information. These sensors produce a 3D point cloud and require new methods for estimating the DBH based on algorithms that automatically detect geometric shapes [33–39]. The algorithms for estimating the DBH fit the transversal section of the tree to a geometric figure: a circle, a cylinder [40], or polygons.

The aims of this study are: (1) to provide an effective method for compiling tree inventories in historic gardens when there is a weak GNSS signal; and (2) to assess the precision of the DBH estimates calculated from three fitting algorithms (RANSAC, Monte Carlo, and Optimal Circle) compared to the reference measurements taken in the field. The reference sample used was 378 trees from the Island Garden (a historic garden and UNESCO World Heritage site and a part of the officially designated cultural landscape of Aranjuez).

## 2. Materials and Methods

### 2.1. Case Study Description

The study took place in the Island Garden (Figure 1), a space located near the Royal Palace in the municipality of Aranjuez (3.36° W, 40.02° N), Madrid (Spain) in the spring of 2020. The Island Garden is an example of an Italian/Flemish renaissance garden and was declared an Asset of Cultural Interest (BIC) in 1932 in the Historic Garden category. It was designated a UNESCO World Heritage site in 2001 [41] and is part of the official cultural landscape of Aranjuez (Madrid, Spain).

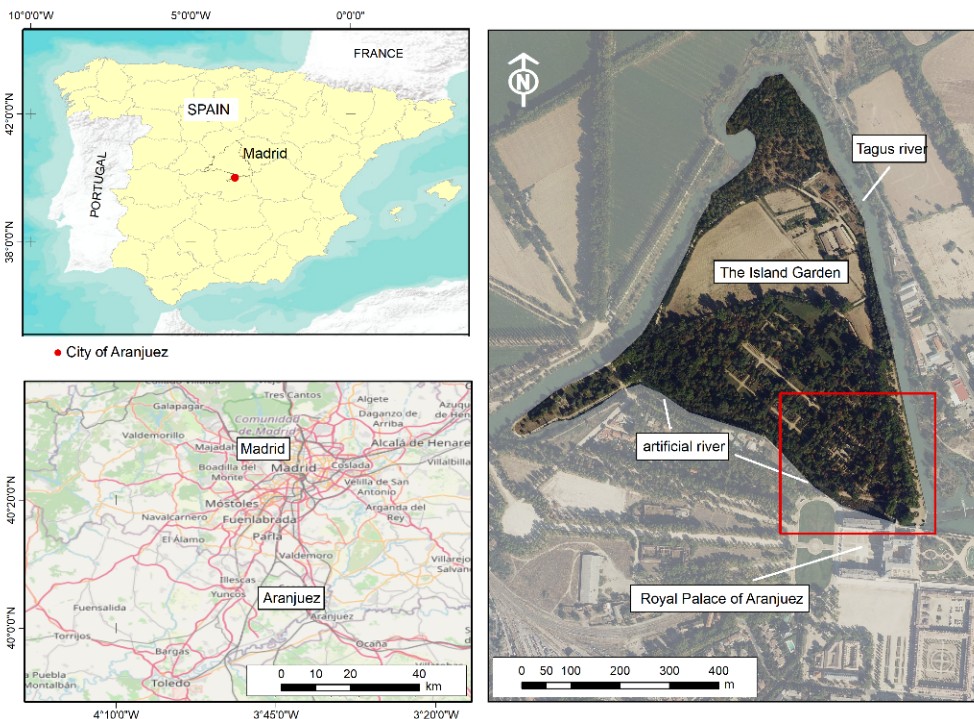

**Figure 1.** The Island Garden is located to the north of the Royal Palace of Aranjuez (Madrid) and forms part of the cultural landscape of Aranjuez officially designated by the UNESCO [42]. The name of Island derives from its location between the Tagus River and an artificial estuary built in the southern part of the garden. The red square denotes the study area. © OpenStreetMap.

The Island Garden is a flat space bordered by the Tagus River and the artificial estuary, and contains various vegetation elements such as rows and groups of trees, copses, flowerbeds, and box hedging (*Buxus sempervirens* L.). It also has a number of ornamental features including fountains, furniture (stone benches), statues, and stone urns. These elements are linked by other types of constructions such as paths, bridges, an aviary, an obelisk, and an arbour. The study area covers 2.38 ha and is located to the southeast of the Island Garden (Figure 1).

The most characteristic tree in the Island Garden is the London plane (*Platanus hispanica* Münchh.). Several of the tree species it contains are classified as singular in the Madrid Region [43], including *Magnolia grandiflora* L. and *Platanus hispanica* Münchh. A variety of herbaceous plants grow in the flowerbeds in the squares, including particularly acanthus (*Acanthus mollis* L.), lilies (*Lilium* L.), zinnias (*Zinnia* L.), dahlias (*Dahlia* Cav.), petunias (*Petunia* ˣ *hybrida* hort), impatiens (*Cardamine impatiens* L.), and hydrangea (*Hydrangea macrophylla* Ser.). The historic garden is managed and maintained by the National Heritage Institution (Patrimonio Nacional) through its Department of Real Estate and Natural Environment [44].

### 2.2. Materials

A Zeb Revo handheld mobile laser scanner developed by GeoSLAM Ltd. (UK) [45] was used to obtain the point cloud. It consists of a laser scanner, a data recorder, a camera, an inertial measurement unit (IMU), and a battery. With a weight of 1 kg (3.5 kg with batteries), this handheld laser scanner is light and portable (86 × 113 × 287 mm) and has a data collection speed of 43,200 points/s, with a relative precision of 2–3 cm (when measurements are only distances, not coordinates positions) and an absolute precision of 3–30 cm (when the measurements are absolute point coordinates), a scanner line speed of 100 Hz, and a maximum range of 30 metres, or 15 to 20 metres, depending on the environment where the 3D point cloud is to be created. MLS technology without GNSS is ideal for this application. A laser profiler measures the profiles with a laser beam launched

from the equipment's rotor, and the device's twists and turns are monitored by means of an inertial system that registers the variations. Both measurements form a 3D model along the path taken by the operator, and errors due to deviations in the inertial navigation system (INS) are compensated by their integration in the point cloud [46,47]. The need to avoid GNSS in areas with little satellite cover, as in the case of trees under the canopy, presents an opportunity to use MLS technology [48], which is currently being applied with success in similar environments where GNSS is unsuitable, such as urban areas, building interiors, underground areas, and forest inventories [26].

Due to the design and type of the historic garden, with very large trees and a dense canopy, the MLS sensor offers certain very interesting features for generating the 3D model, thanks to its mobility. The use of SLAM technology in this type of space is important, as it does not depend on satellite positioning and makes it possible to operate in places with a weak, low-quality satellite signal (Figure 2).

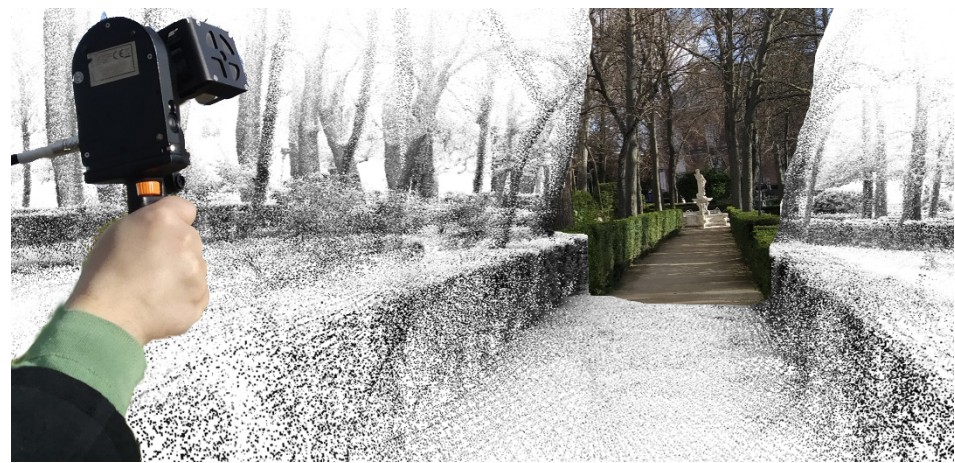

**Figure 2.** The MLS sensor contributed effectively to acquiring the 3D point cloud. A new topographic map has been generated to meet the need to update the tree inventory in a historic garden. This is an agile and powerful tool for the management and maintenance of a protected space.

The basic vector map was supplied by the Spanish Government Office in Aranjuez for the Department of Real Estate and Natural Environment at the National Heritage Institution (Patrimonio Nacional) and was used to geolocate the tree inventory, and as a base map for the analysis.

The DBH of all the trees in the surveyed area (378 trees) was measured with a caliper with a precision of 1 mm. The trees with a diameter larger than the caliper maximum (60 cm) were measured using a diameter tape. A base vector map provided by Patrimonio Nacional was used to geolocate the trees whose DBH was measured with a caliper. The vector map was an AutoCAD file (.dwg) in the ETRS89 UTM Zone 30N projection (EPSG: 25830), with several layers that included information on the location of walls, benches, statues, fountains, signage, hedges, trees, and shrubs. The tree locations were exported to a point shapefile (.shp) and the tree DBH were added to this shapefile as a field in the attribute table. The caliper measurements were recorded in the attribute table of the tree shapefile. Information on the species and status of each tree was also recorded. The shapefile is available in the Supplementary Materials. The MLS acquisition and DBH measurements were made on 6 March 2020.

### 2.3. Methods

Key parameters, such as drift and scan width, point density, and the size of the files to be handled for this type of equipment, were considered before capturing the data. The definition of the values was analysed for each one, taking into account the fact that the work took place outside, which makes the scanning process more complex. The parameters

selected were: distances of 400 m or less, scan widths between passes of no more than 20 m, common areas in each circuit, and scan times with a close of almost 5 min.

The methodology (Figure 3) includes a preliminary fieldwork phase with MLS, consisting of initialising the sensor, following the prescribed path, and returning to the same starting point, then closing the trajectories in order to correct the bias produced by the INS during capture. Due to the cumulative error during long recording times, it was decided to reduce the distance of the closed paths in order to minimise the influence of the drift error of the equipment on the accuracy of the captured data. Six different paths were established to cover the work area (Figure 4). The overlaps allowed the layers to be integrated with the adjoining trajectories and ensured that the system bias was controlled by means of points that are common to all of them. Different geometries were defined in the trajectories in order to analyse their correctness according to the shape of the park. The total duration of the survey by MLS was 27 min 37 s, and a 3D point cloud was obtained for each path. Table 1 shows the distance covered and the 3D points surveyed in each path.

**Table 1.** Results of the 3D point cloud survey by Zeb Revo laser scanner.

| Path | Distance (m) | Points | Time |
|---|---|---|---|
| 1 | 298 | 5,420,396 | 5 min 11 s |
| 2 | 300 | 5,325,954 | 5 min 27 s |
| 3 | 378 | 3,072,892 | 3 min 14 s |
| 4 | 186 | 5,052,089 | 4 min 48 s |
| 5 | 156 | 3,829,763 | 3 min 49 s |
| 6 | 272 | 5,383,714 | 5 min 08 s |
| Total | 1590 | 28,084,808 | 27 min 37 s |

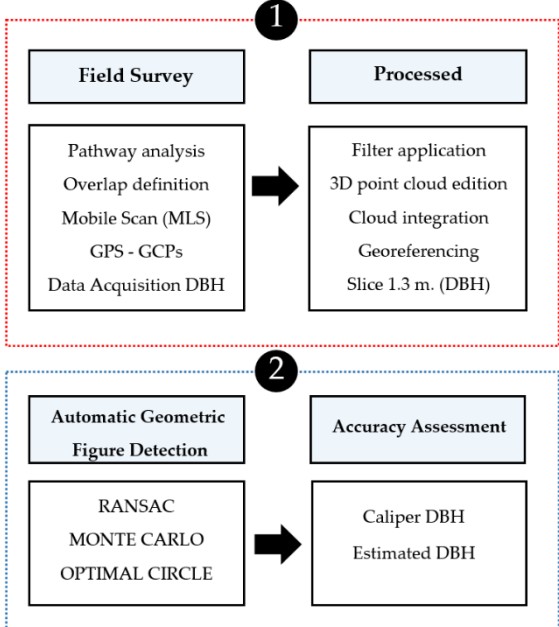

**Figure 3.** Methodological diagram of the study consisted of a preliminary fieldwork phase and the subsequent processing of the different 3D point clouds, and a second phase to assess the accuracy of DBH estimation.

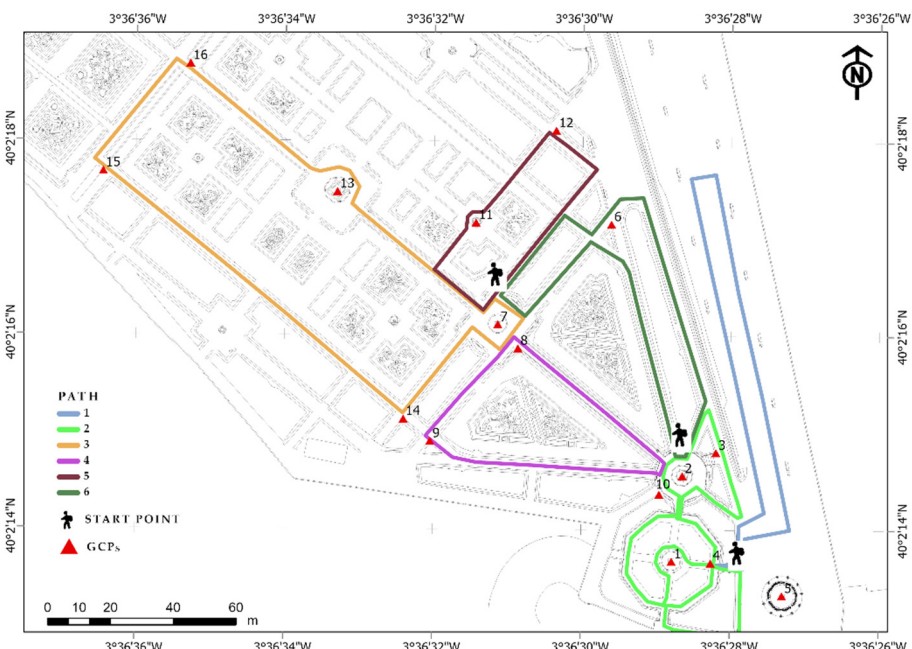

**Figure 4.** Location of the GCPs and layout of the six paths in the study zone. Three starting points were planned: one for paths 1 and 2; one for paths 3 and 5; and one for paths 4 and 6. There are several overlap areas to ensure the optimum integration of the different 3D point clouds.

The coordinates of the 16 ground control points (GCP) were obtained by GNSS post-processing techniques to improve the satellite measurements under the trees. These points are easy-to-scan, 40-cm squares with a black and white checkerboard pattern, used as predesigned marks, to indicate the centre in GNSS measurements. The locations of the GCP points were plotted and distributed so as to cover the entire study area, and as required to geo-reference the different 3D point clouds (Figure 3). The equipment used was a Leica Geosystems GX1230 GG, and the data were processed with the Leica Geo Office [49] software, with a relative average accuracy of more than 0.02 m at all points.

The 3D point cloud for each path was downloaded from the equipment and automatically processed by the ZEB REVO postprocessing software to reprocess inertials, point clouds, and paths, generating a bin extension file with the drifts corrected and the internal crossed trajectories adjusted independently for each path. The point clouds were imported in FARO Scene© and the points were automatically recorded cloud-to-cloud, progressively analysing different subclusters in an iterative process that ultimately determines the best relative position of the clouds. Once a single cloud had been determined with these six trajectories, the GCPs in the point cloud were identified; care was taken during scanning to define these points for their subsequent identification. Finally, the point cloud was projected into the same cartographic system as the tree shapefile (UTM Zone 30). All processing was done in CloudCompare [50], a free OpenCode software for processing 3D point clouds obtained by laser scanner. The initial output of the integration of all the 3D point clouds is a 3D model of the study area which can be used for obtaining different views, sections, 3D itineraries etc., all of which are very useful for the management and modelling of historic gardens (Figure 5).

A significant problem in the use of the MLS Zeb Revo sensor is the noise captured in the 3D cloud [51]. In order to eliminate this noise and optimise the resulting 3D model of the grouping of the six paths, the 3D cloud was subjected to several filtering and editing processes. The first filter used in the CloudCompare software was the CSF filter: This is a plugin that applies an algorithm to detect the ground in the point cloud from a series of parameters (Figure 6). After assessing different parameters, a resolution of 0.5 m and a classification threshold of 0.3 m were chosen. The 3D model was also edited to eliminate any points observed by the sensor that had no value to the study, specifically the presence

of points captured on people who were walking through the garden at the time of the observation, and points above the cut-off at a certain height. The filtered 3D point model was reduced to 12,124,000 points.

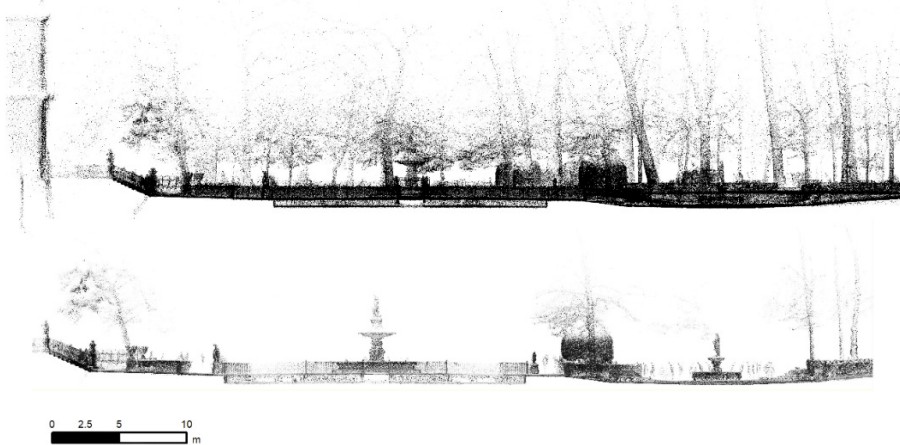

**Figure 5.** Different sections of the Island Garden showing fountains, furniture, and trees, and allowing the measurement of distances, height of first branches, etc. This is a useful tool for operators responsible for the management and maintenance of the historic garden.

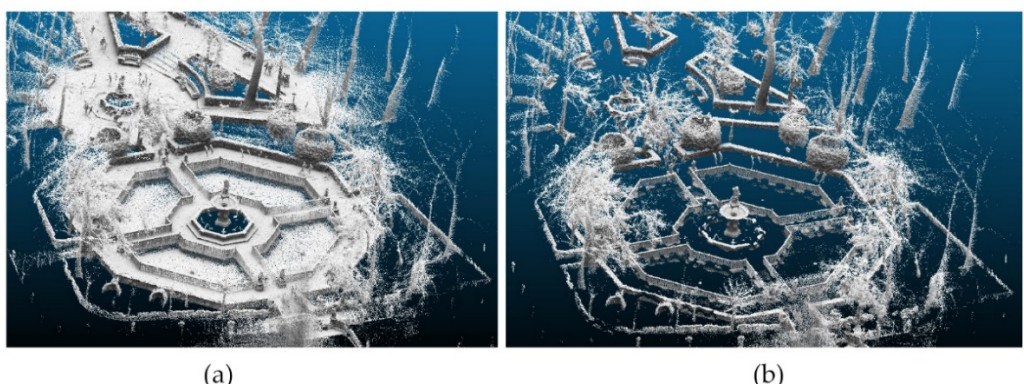

(a)                                                               (b)

**Figure 6.** (**a**) A zone of the original 3D point cloud and (**b**) the application of the CSF filter plugin for ground detection, which optimises the 3D point cloud for further processing.

In order to estimate the DBH of the trees, a horizontal slice was extracted from the filtered point cloud. The slice had a thickness of 20 cm and consisted of points between 1.30 and 1.50 m above the mean elevation of the GCPs, which was roughly the same for all GCPs, as the terrain is completely flat. A 20 cm slice was estimated to obtain sufficient points in the point cloud and to ensure that the algorithms could estimate the corresponding DBH (Figure 7). Any points not corresponding to trees were eliminated manually. The tree cross-sections in the slice were matched with the records in the tree shapefile to later add the three automated DBH estimates to the attribute table. For this, the reported position of the trees in the point shapefile were manually adjusted to make them coincide with the centroid of the cross-sections in the cleaned-up slice.

Figure 7 shows the slice for estimating the DBH in different trees. The main problem in specimens with a DBH of less than 50 cm (Figure 8a,b) was the presence of noise, which hindered the fitting of the geometric shape, and hence the estimation of the DBH. In specimens with an irregular morphology (Figure 8c,d), some or all of the geometric shape fitting algorithms were unable to find a circle or cylinder in the cross-sections.

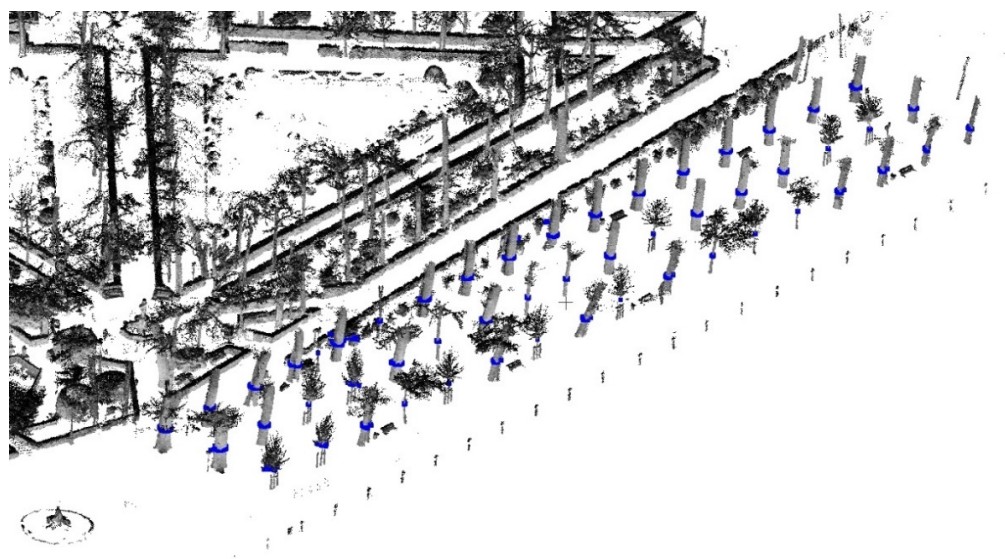

**Figure 7.** Section of the point cloud on path no. 1 shown in blue circles at a height of 1.3 m above the ground.

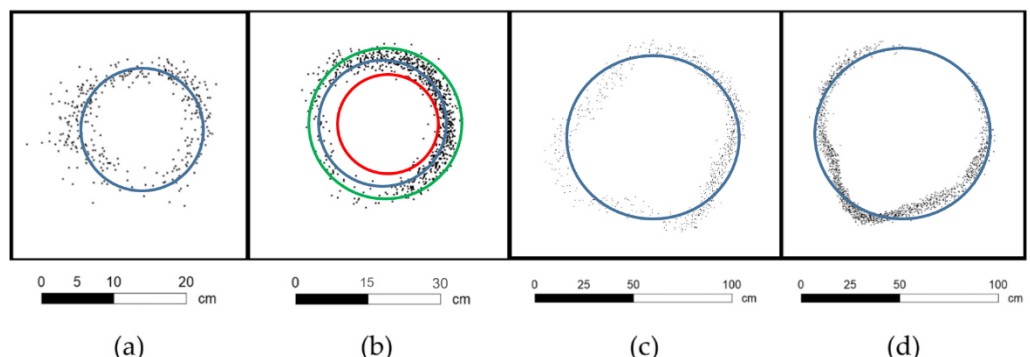

**Figure 8.** Cross-section of specimens in the different intervals ([0–0.25], [0.25–0.50], [0.50–1.00], [1.00–... ]): in (**a**) tree 171 with a DBH of 18.3 cm, (**b**) tree 212 with a DBH of 26.0 cm, (**c**) tree 173 with a DBH of 99.6 cm, and (**d**) tree 228 with a DBH of 128.6 cm. (**b**) also shows the biggest differences on the DBH estimation: blue circle corresponds with RANSAC algorithm (26 cm), red is OC (22 cm), and MCM is green circle (32 cm).

The accuracy of the DBH estimates made with each of the three methods was assessed by the root mean squared error (RMSE), the coefficient of determination ($R^2$), and the bias [19,20]:

$$RMSE = \sqrt{\sum_{i=1}^{n} \frac{(DBH_i - DBH_{ref\,i})^2}{n}} \tag{1}$$

$$R^2 = \frac{\sum_{i=1}^{n}(DBH_i - \overline{DBH_i})^2}{\sum_{i=1}^{n}(DBH_{ref\,i} - \overline{DBH_i})^2} \tag{2}$$

$$Bias = \sum_{i=1}^{n} \frac{(DBH_i - DBH_{ref\,i})}{n} \tag{3}$$

where n is the number of trees with at least one DBH estimate (258), $DBH_{ref\,i}$ is the reference value measured in the field, and $DBH_i$ is the estimated measurement.

In addition, various analyses were conducted to assess how error magnitude and bias vary with the DBH.

## 3. Results

As in the study by Ryding et al. [52], the DBH estimation was more complicated in trees with a DBH of less than 10 cm. In our study there were 120 (32%) cases in which we could not use geometric shape detection algorithms. These 120 specimens correspond either to trees with a DBH below 10 cm, or to trees located at the edges of the surveyed area, and have been exclude from the DBH analysis (out of the 120 excluded trees, 41 did not yield an estimate in the RANSAC algorithm, 88 in the MCM algorithm, and 93 in the OC). All the trees excluded from this study share the common condition that one or more of the algorithms assessed was unable to estimate their DBH. The final number of trees used to assess the algorithms was 258 (Figure 9).

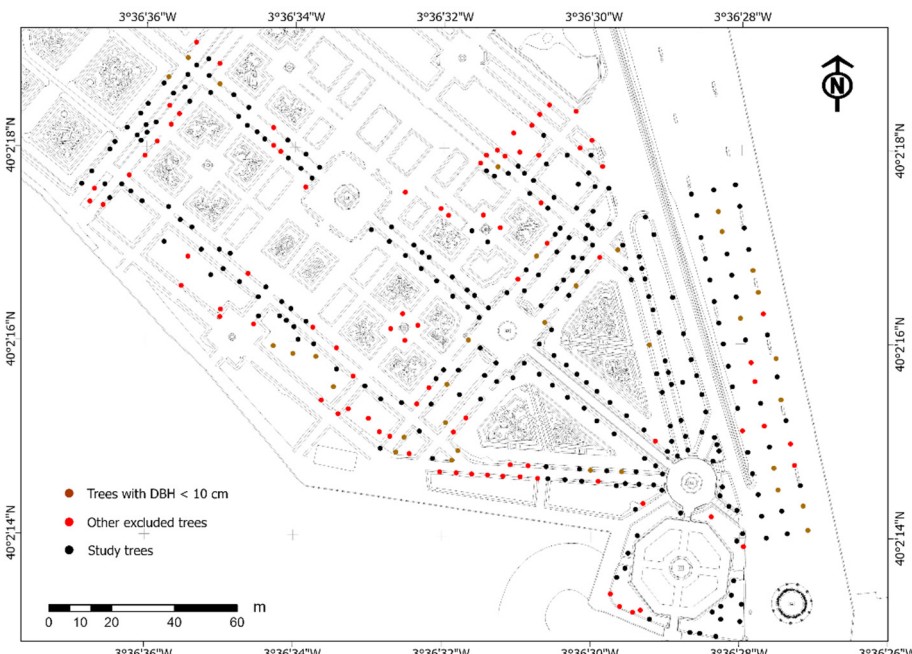

**Figure 9.** Trees used to analyse the various fitting algorithms and assess the accuracies of the DBH estimates are shown in black. Trees shown in red are excluded from the DBH analysis because all or some of the geometric shape fitting algorithms were unable to find a circle or cylinder in those cross-sections. Trees shown in brown are excluded from the DBH analysis because they are trees with a DBH of below 10 cm.

Three different automatic geometric shape detection algorithms were tested for estimating the DBH from the 3D point cloud. These were the random sample consensus (RANSAC) [53], the Monte Carlo method (MCM) [54], and Optimal Circle (OC) [54], which are suitable for detecting geometric objects in 3D point clouds.

The RANSAC shape detection plugin is an algorithm that fits a geometric shape such as a line, plane, circle, sphere, or cylinder from a 3D point cloud and from different parameters to the model created with geometric algebra [55]. It is an iterative method that extracts primitive shapes from a minimum number of points and then tests them against all the points in order to estimate the one most similar to the point cloud [24,36].

The MCM and OC are algorithms for refining and fitting circles; both detect the two furthest points, analyse possible circles in the transversal section of the point cloud, and generate random changes until the required precision is reached. These two algorithms are included in the Dendrocloud software [54]. The MCM and OC user parameters were set as follows: count limit, 1000; RMS limit, 0.01; and the initial approximation as automatic. A RANSAC algorithm from CloudCompare [50,56] on forest inventories [35,57] was used to perform a third option for the DBH estimation. This algorithm does not allow the computation of 258 trees at once. For this reason, the authors have implemented their own RANSAC algorithm (github.com/GESyP/ransac_gesyp) with the following parame-

ters: maximum distance to primitive, Radious + sigma/2; sampling resolution, 0.05; and overlooking probability, 0.010.

The full set of 378 trees in the study area (Figure 10a) contains a greater number of trees below 0.5 m than the subset of 258 trees for which a DBH estimate was produced (Figure 10b). Figure 10a shows the sample used to assess the accuracy of the three algorithms. These 258 specimens are distributed as follows: 20.16% of specimens have a DBH of <0.25 m., 52.71% have a DBH of between 0.25 and 0.5 m, 22.87 have a DBH of between 0.5 and 1.0 m, and 4.26% have a DBH of over 1.0 m. Figure 10b shows all the 378 trees inventoried in the study area, with the following distribution: 37.57% of specimens have a DBH of <0.25 m, 42.59% have a DBH of between 0.25 and 0.5 m, 16.67% have a DBH of between 0.5 and 1.0 m, and 3.16% have a DBH of over 1.0 m.

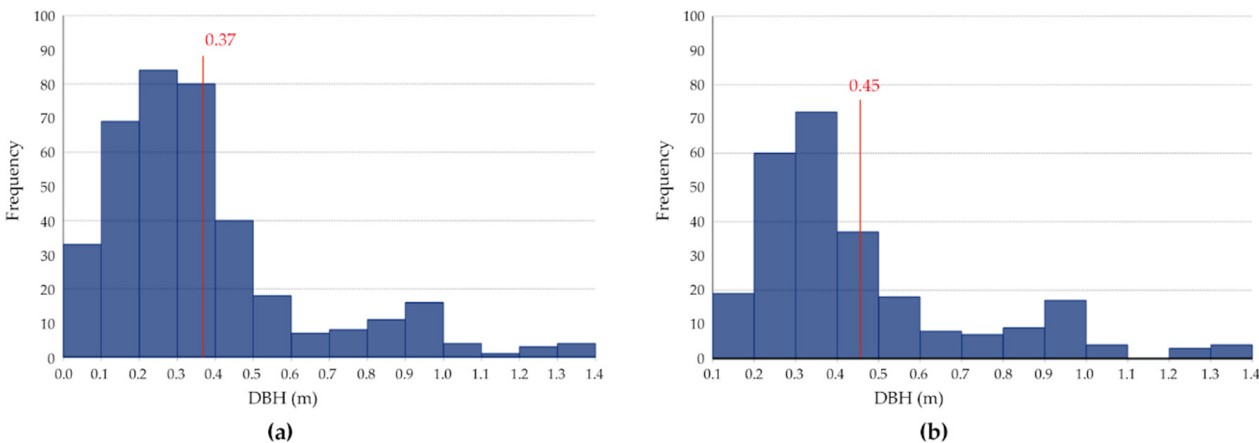

**Figure 10.** Histograms of the field-measured DBH for (**a**) the 378 trees in the study area and (**b**) the subset of 258 trees for which an automated DBH estimate was successfully produced.

Using the reference DBH measured in the field and the 3D point cloud, a primitive geometric shape was fitted to each tree cross-section and its centroid was estimated. This made it possible to update the tree inventory in the study area in the Island Garden. This is a rapid and useful tree inventory method for the management and maintenance of the historic garden.

Following the methodology proposed in the previous section, the first analysis consisted of comparing the reference diameters measured in the field with a tree calliper and estimating the DBH with the automatic calculations made by the three procedures explained above. All trees in the study area had a visible cross-section in the MLS slice, but insufficient points were obtained in 31.7% of the trees to be able to estimate their corresponding DBH.

The OC method has an $R^2$ of 0.9694 and an RMSE of 0.052 m, which corresponds to a relative error of 11.5% (relative to the mean DBH) (Figure 11). The MCM method has an $R^2$ of 0.9634 and a similar RMSE of 0.053 m, whereas the $R^2$ of the RANSAC is 0.9602 and the RMSE is similar with 0.060 m, representing an error of 13%.

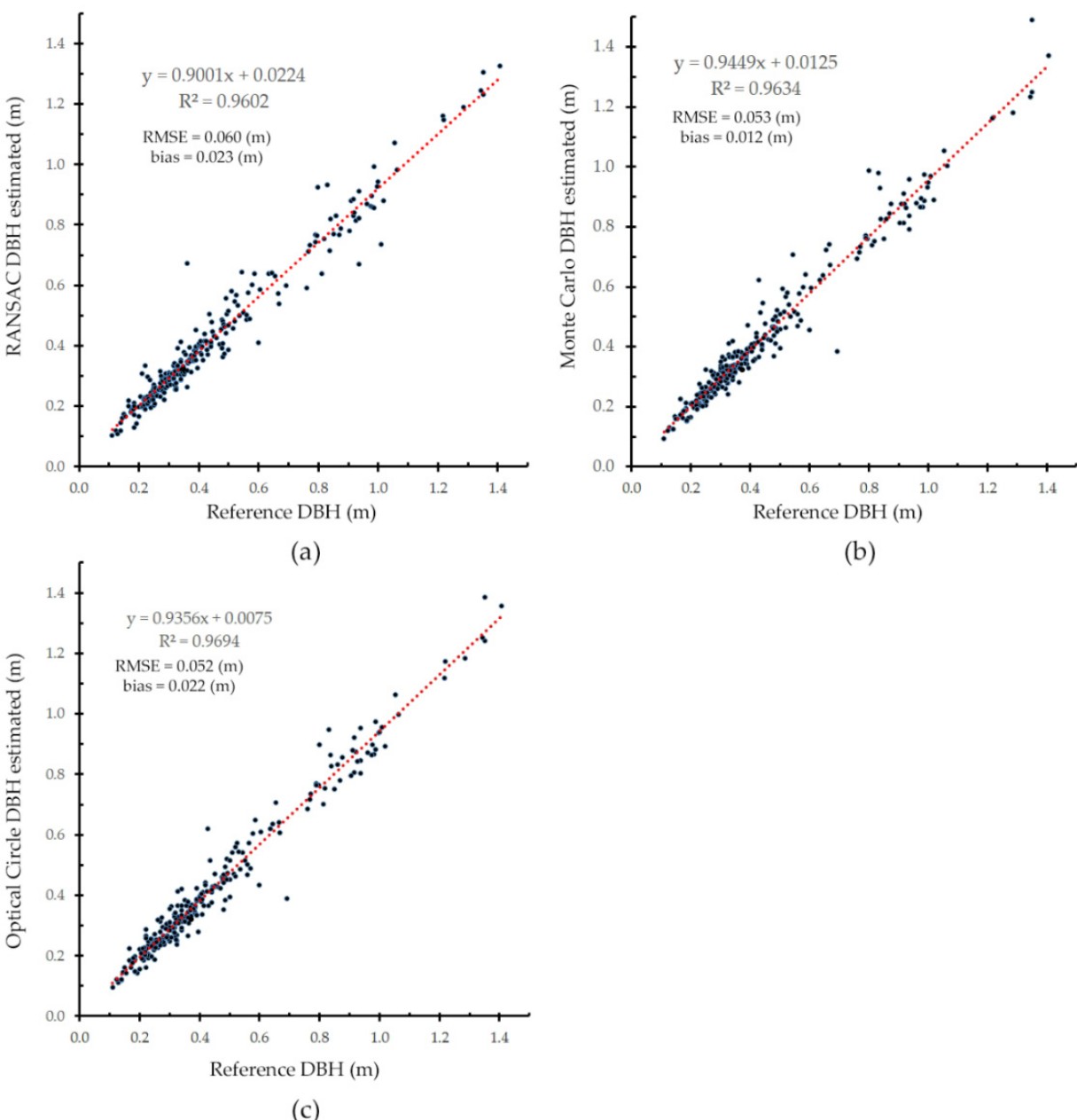

**Figure 11.** Dispersion of DBH estimated with the three algorithms and the reference values measured in the field (m). (**a**) The RANSAC fitting was used, (**b**) as was the MCM and (**c**) OC.

As shown in Figure 11, the error in the automatic estimation of the DBH of trees is slightly proportional to the diameter. Figure 12 shows a box diagram with the dispersion of the errors in different DBH intervals: <0.25 m, 0.25–0.5 cm, 0.5–1 m, and >1 m. The MCM and OC methods produced a similar result, although the first has less variability, so its DBH are more homogeneous. The RANSAC estimates are more distant for the DBH in the over 1.0 m range, and the MCM algorithm presents isolated points.

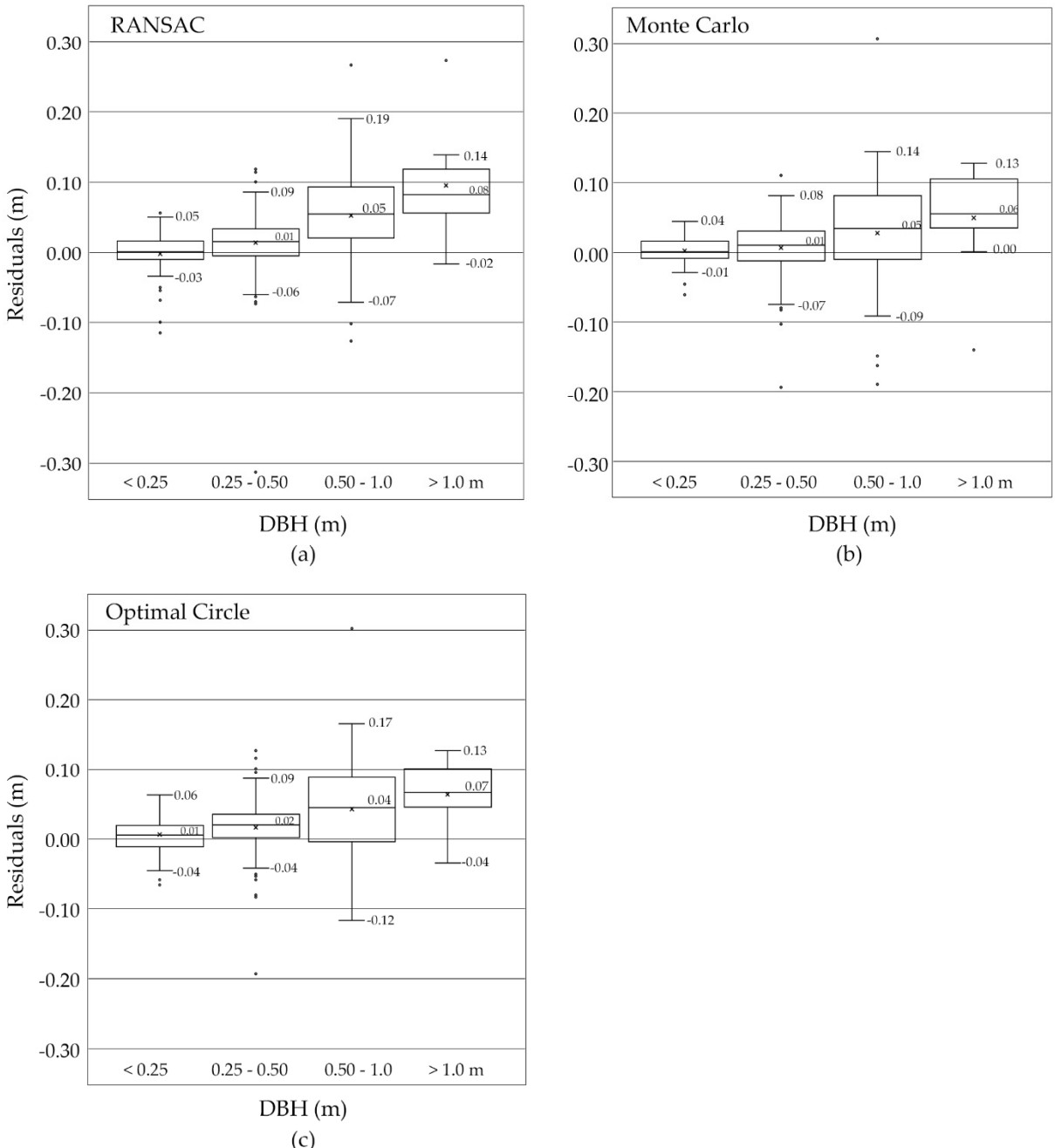

**Figure 12.** Distribution of DBH residuals of each algorithm by intervals. In (**a**) the RANSAC fitting was used, in (**b**) the MCM, and in (**c**) the OC. Where asterisks are the mean value and circles are outliers values.

The third assessment focuses on the determination of the RMSE and the bias in the three methods, with similar outcomes to the previous ones. The error in the automatic determination of the tree diameters is proportional to the diameter. The RMSE of the reference values compared to the estimates of the RANSAC algorithm is 6.04 cm, with a bias of 2.26 cm. The OC algorithm has an RMSE of 5.25 cm and a bias of 2.15 cm, and the best results are achieved with the MCM, which has an RMSE of 5.31 cm and a bias of 1.23 cm.

To summarise the results, the accuracy of the DBH can be said to follow the same trend based on the intervals analysed in the three algorithms (Figure 13). Trees with a DBH of <0.25 m have more accurate estimates, and the error in the accuracy of the estimate for DBHs over 0.50 cm is in the range of 7 to 12 cm.

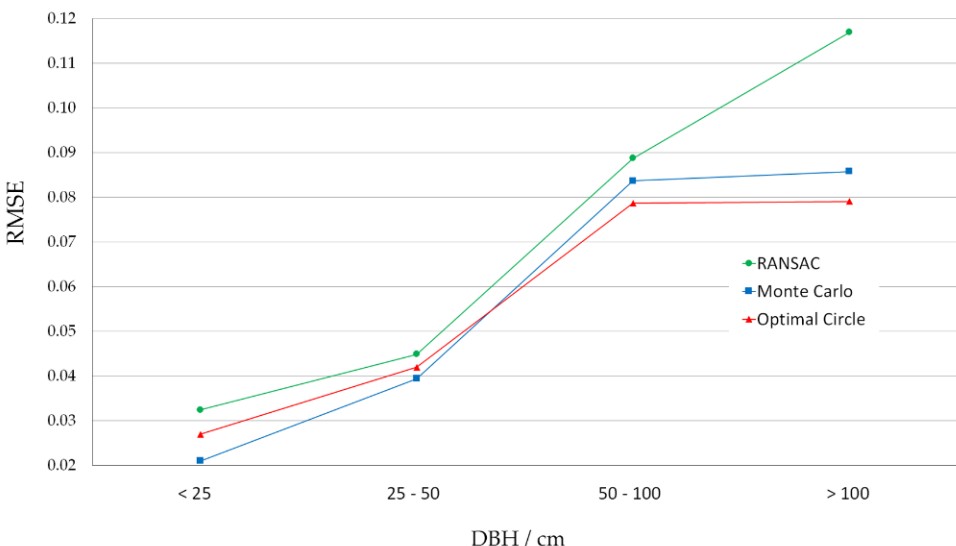

**Figure 13.** RMSE in the various intervals in each method analysed. Specimens with DBH under 25 cm have an RMSE of less than 3 cm. After a DBH of over 50 cm, the RMSE is over 8 cm.

Table 2 shows the RMSE and bias obtained by the different algorithms, both by the DBH interval and in total. The three algorithms assessed produced good results when the estimated DBH was below 50 cm. The best estimate was made by the Monte Carlo algorithm with specimens with a DBH of less than 25 cm (with an RMSE of 2.10 cm and a bias of 0.21 cm).

**Table 2.** RMSE and bias in the estimated accuracy of DBH in the algorithms by intervals.

| | RANSAC | | Monte Carlo | | Optimal Circle | |
|---|---|---|---|---|---|---|
| **DBH (m)** | **RMSE (cm)** | **BIAS (cm)** | **RMSE (cm)** | **BIAS (cm)** | **RMSE (cm)** | **BIAS (cm)** |
| <0.25 m | 3.24 | −0.22 | 2.10 | 0.21 | 2.70 | 0.54 |
| 0.25–0.5 m | 4.48 | 1.34 | 3.94 | 0.65 | 4.20 | 1.54 |
| 0.5–1.0 m | 8.87 | 5.22 | 8.37 | 2.75 | 7.86 | 4.19 |
| >1.0 m | 11.68 | 9.47 | 8.58 | 4.94 | 7.90 | 6.31 |
| total | 6.04 | 2.26 | 5.31 | 1.23 | 5.25 | 2.15 |

## 4. Discussion

Although the MLS generated returns for all the trees in the area surveyed (378 specimens), insufficient points were obtained to estimate the corresponding DBH for 31.7% of the trees. These cases correspond to trees with a DBH of less than 10 cm (8.73%) and to trees that have insufficient points in the slice (23.27%), making it impossible for the algorithms to correctly estimate their DBH. These trees (32%–120 specimens) were not included in the DBH portion of the study. One problem we faced in our methodology was that the actual design of a historic garden (the arrangement of the paths and the presence of hedges, ornamental fountains, garden benches, signage, etc.) requires prior planning when surveying via MLS, and the subsequent editing and filtering of the points in the resulting 3D model. To increase the number of points around the cross-section of trees and to increase the proportion of trees with an automated DBH estimate, the MLS paths could be designed following a grid of transversal paths. As in the study by Bauwens [14] on the differences between the use of TLS and MLS in forest inventories, one drawback of MLS is its poor range above the canopy in the historic garden, which was between 13 and 20 m. Even though our focus was not on tree height, this represents a handicap for obtaining a complete inventory that includes tree height.

The DBH estimated with the algorithms in the study was satisfactory. The three procedures correctly calculated the tree diameter with a mean error of 5.53 cm and an overestimation bias of 1.88 cm. An analysis of each method individually shows that the MCM and OC algorithms obtained similar results (a difference of 0.6 mm in their

mean), although the bias in the MCM was half than that in the OC. The RANSAC had just a 7.3 mm larger RMSE than the MCM, although it also had 80% more overestimation bias.

The results are similar to those obtained by Koren [17], who used TLS and included 157 European beeches (*Fagus sylvatica* L.) in 50 × 50 m fields. They estimated the DBH from several fitting methods, and, as in our study, the OC method proved to be the most accurate in terms of the RMSE.

There are other comparable works that assess the different methods of estimating the DBH. For example, Forsman et al. [33] obtained an RMSE of 3.7 cm, using MLS as in our work, in 20 × 20 m fields and with a 2D projected circle fit. Gollob [15] reported an RMSE of 4.18 cm, using hexagons and not circles as the best fit for estimating the DBH. By using MLS and the RANSAC algorithm, Cernava [57] obtained an RMSE of 5.57 cm for the estimated DBH, very similar to the 6.04 in our study. Finally, Liang et al. [58] achieved better results than ours with an RMSE of 2.36 cm, using MLS in a study of Scots pine (*Pinus silvestris* L.) in 63 trees, and with a DBH estimate using a circle fitting method.

These algorithms estimate the geometric shape of the point cloud that most closely fits the tree bark. Factors influencing the differences include the tree's roughness and the irregular trunk forms in adult specimens, which affects the fitting results in the case of circles.

There are currently differences in survey costs compared to other sensors, such as TLS. An area of 2.38 ha was surveyed in less than 30 min. Liang et al. [29] used a backpack-mounted MLS to make a 3D model of 2000 m$^2$ in a time of two minutes. Some studies have compared the TLS and MLS system, and the field time for surveying trees with MLS is 12 times faster than with TLS [52], and up to 40 times faster in areas of difficult topography [59]. The MLS Zeb Revo equipment has an approximate cost of $30,000, lower than that of other technologies such as TLS ($40,000–100,000 depending on the manufacturer).

## 5. Conclusions

This study used an MLS based on SLAM to compile a tree inventory in a historic garden, and assessed the accuracy of the DBH estimates derived from three fitting algorithms.

The methodology described proved to be an effective means of detecting all trees and estimating the DBH of most of them in a space with a series of specific characteristics, such as the presence of ornamental fountains, furniture, signage, and the alignments of hedges or plant species that hinder data collection. The different paths followed must be correctly planned in order to be able to edit, and subsequently optimise, the 3D point cloud. The reliability and precision of the tree inventory in historic gardens improves their management and maintenance.

It took 12 min to compile an inventory of the trees in one hectare with MLS, after correctly planning the paths to follow. The MCM fitting method provided slightly better results than the others for estimating the DBH, with an RMSE of 5.31 cm and an overestimation bias of 1.23 cm.

In the coming years, the use of MLS can be expected to become widespread in tree inventories, as it obtains 3D data quickly and effectively, although new routines will need to be developed for the management and accuracy of SLAM technologies under a large tree canopy.

**Supplementary Materials:** The following are available online at https://www.mdpi.com/article/10.3390/f12081013/s1, File S1: Shapefile.

**Author Contributions:** Conceptualization, E.P.-M., S.L.-C.M. and T.H.-T.; formal analysis, E.P.-M. and S.L.-C.M.; investigation, T.H.-T., J.A.d.M., A.E.-C. and M.A.P.-S.; methodology, E.P.-M., J.A.d.M. and A.E.-C.; software, S.L.-C.M. and M.A.P.-S.; visualization, E.P.-M. All the authors contributed to writing and original draft preparation. All authors have read and agreed to the published version of the manuscript.

**Funding:** This research was funded as a part of the research tasks (data collection) under the SENSE-SCAPES R&D Project, Ministerio de Economía y Competitividad, HAR2015-64762-P. Dissemination

and additional research tasks (data processing) was funded by the LABPA-CM R&D Program, European Social Fund and the Community of Madrid, H2019/HUM-5692.

**Data Availability Statement:** The data presented in this study are openly available in Zenodo at https://doi.org/10.5281/zenodo.5136935.

**Acknowledgments:** We would like to thank the Department of Real Estate and Natural Environment of the National Heritage Institution-Aranjuez Office (https://www.patrimonionacional.es) and the company GEOAVANCE S.L. for its cooperation. This paper has been revised/translated by Pru Brooke-Turner.

**Conflicts of Interest:** The authors declare no conflict of interest.

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
