# Peer review of "Assessment of Tree Diameter Estimation Methods from Mobile Laser Scanning in a Historic Garden"

_forests, doi:10.3390/f12081013_

Round 1

Reviewer 1 Report

Dear authors,

I carefully read your new manuscript and I have to say, that now everything is much better. Also English is fine a I had no problem to understand the text (it is a really contrast comparing your previous version).

I have only a few notes:

Line 227 - did you want to write "shapefile"?

Line 444 - L. is missing in the pine latin name

Have a good luck with publicating this article!

Author Response

First we would like to thank you for the further opportunity you have given us to improve the manuscript in this second resubmision and also in our first time. Your suggestions have really change for the bes tour manuscript. We have included your suggestions as follows:

I have only a few notes:

Line 227 - did you want to write "shapefile"?

Modification included as suggested

Line 444 - L. is missing in the pine latin name

Modification included as suggested

Br,

Reviewer 2 Report

The authors present a study on using MLS data to measure DBH, and compare with bias of three methods. I think it is a good paper for forestry and Lidar measurement, but it is unfinished article in the present form.

In Abstract, LINE 24-25. In this study, higher DBH gets higher RMSE and bias, so this paper's best result at how many DBH?

In Introduction, the fitting algorithms are most important parts in this manuscript, to introduce often see method and your method, and why you are choosing those three methods.

LINE 57-57. It's a challenge with TLS, I think you need to cite more reference.

LINE 139. From my experience, the large size caliper is 0.6 m, why you could measure DBH more than 0.6 m, could you attach the instrument model? In addition, the tree trunk isn't a circularity, when people measure different position of trunk at same height, that have different diameter.

LINE 169. The workflow must be to explain things step by step.

Methods and Results both are necessary to separate to chapter, section, paragraph, and title.

The font size and marks are very small in every Figures, to print this manuscript and check text is clean.

Could you show some examples about your fitting results on point cloud of three methods?

Figure 10, we can saw have big bias at three methods, to show a new figure and to draw maximum bias on point cloud of trunk, and why?

LINE 405-407, this paragraph moves to Introduction or Materials, and figure 13 moves to Materials. As I see it, this paragraph is not discussion.

Author Response

First we would like to thank you for the further opportunity you have given us to improve the manuscript. We hope that the changes we have introduced will warrant a more favourable and definitive assessment from you. We believe the paper has been much improved thanks to your suggestions.

In Abstract, LINE 24-25. In this study, higher DBH gets higher RMSE and bias, so this paper's best result at how many DBH?

We have included a complete reference of how many DBH were estimated in the best result of the study, changing the required line 24-25 as follows:

The best results were obtained with the Monte Carlo fitting algorithm, which was able to estimate the DBH of 77% of the 378 trees in the study, with a root mean squared error (RMSE) of 5.31 cm and a bias of 1.23 cm.”

In Introduction, the fitting algorithms are most important parts in this manuscript, to introduce often see method and your method, and why you are choosing those three methods

During the process of reviewing the state of the art on computer tools in forest inventories [3, 4, 7, 12 - 18, 20 - 25, 30 - 37], the authors analyzed the different methods used, finding these three as the most used for DBH estimation, mainly using TLS, so an evaluation of methods with MLS was necessary.

LINE 57-57. It's a challenge with TLS, I think you need to cite more reference.

We have studied several papers of TLS because as you suggest, this method is very used, so our challenge is to change the strategy to dinamyc methods such as MLS. We have reference TLS in 11, 12, 17, 19. We have updated and added new studies related to TLS tools in the area of forest inventory.

Li, L.; Mu, X.; Soma, M.; Wan, P.; Qi, J.; Hu, R.; Zhang, W.; Tong, Y.; Yan, G. An Iterative-Mode Scan Design of Terrestrial Laser Scanning in Forests for Minimizing Occlusion Effects. Ieee Transactions on Geoscience and Remote Sensing 2021, 59, 3547-3566, doi:10.1109/tgrs.2020.3018643.

Kukenbrink, D.; Gardi, O.; Morsdorf, F.; Thurig, E.; Schellenberger, A.; Mathys, L. Above-ground biomass references for urban trees from terrestrial laser scanning data. Annals of botany 2021, doi:10.1093/aob/mcab002.

Xia, S.; Chen, D.; Peethambaran, J.; Wang, P.; Xu, S. Point Cloud Inversion: A Novel Approach for the Localization of Trees in Forests from TLS Data. Remote Sensing 2021, 13, doi:10.3390/rs13030338.

LINE 139. From my experience, the large size caliper is 0.6 m, why you could measure DBH more than 0.6 m, could you attach the instrument model? In addition, the tree trunk isn't a circularity, when people measure different position of trunk at same height, that have different diameter.

The DBH measurement in each tree, using the caliper, was made at a height from the ground of 1.30 m (see line 75). For trunks greater than 0.6 m in diameter, a tape measure was used to measure the circumference and subsequently obtain the DBH of the tree. In those trees that presented wounded protuberances, hollows, branches, etc. at breast height, the measurement was obtained just above the irregular point, where the irregular shape did not affect the trunk.

LINEA 169. El flujo de trabajo debe ser para explicar las cosas paso a paso.

Modification included as suggested. The modification is: “Methodological diagram of the study consists of a preliminary fieldwork phase and the subsequent processing of the different 3D point clouds; and a second phase to assess the accuracy of DBH estimation”.

Methods and Results both are necessary to separate to chapter, section, paragraph, and title

Modification included as suggested

The font size and marks are very small in every Figures, to print this manuscript and check text is clean.

We have modified as suggested, printed and reviewed those figures where Font sizes and marks were not readeable. We have included new modified figures 4, 9 and 12.

Could you show some examples about your fitting results on point cloud of three methods?

Figure 8 has been changed to show the point cloud and the estimated diameter. We have included the best diameter estimation and an example of the bigger difference with the three DBH algorithms.

In the same way we have analyzed the request and only 20 trees show a difference of more than 5cm. We have included your suggestion about the three method comparison in this figures because it shows the different scale diameters and the

“Figure 8. Cross-section of specimens in the different intervals ([0-0.25], [0.25-0.50], [0.50-1.00], [1.00-…]): in (a) tree 171 with a DBH of 18.3 cm, (b) tree 212 with a DBH of 26.0 cm, (c) tree 173 with a DBH of 99.6 cm and tree 228 with a DBH of 128.6 cm. (b) shows also the biggest differences on the DBH estimation: Blue circle corresponds with RANSAC algorithm (26cm), Red is OC (22cm) and MCM is green circle (32cm).”

Figure 10, we can saw have big bias at three methods, to show a new figure and to draw maximum bias on point cloud of trunk, and why?

In Figure 11, where the rmse and bias of the DBH estimates are included, the units of bias in cm have been corrected, which improves its appreciation.

LINE 405-407, this paragraph moves to Introduction or Materials, and figure 13 moves to Materials. As I see it, this paragraph is not discussion.

Modification included as suggested. We have renamed the figures again.

Br,

This manuscript is a resubmission of an earlier submission. The following is a list of the peer review reports and author responses from that submission.

Round 1

Reviewer 1 Report

I have some minor request. But it is ready to go.

It is very creative to use this Mobile mapping system for historic garden. It is valuable to publish this type of paper.

Please see my comments below.

Line 183-184 Why did you use FARO software. It is better to clarify the reason of this procedure.

Line 247 -> Figure 9(b)

You did not explain much about Figure 9. Please explain more on this. Is it filtering effect, right ?

Figure 12, please show the unit of X and Y in centimeter instead of meter. 

Nice work!

Author Response

I have some minor request. But it is ready to go.

It is very creative to use this Mobile mapping system for historic garden. It is valuable to publish this type of paper.

We are thankful for your appreciate comments that has helped to improve our document.

Please see my comments below.

Line 183-184 Why did you use FARO software. It is better to clarify the reason of this procedure.

The registration technology based on the Zebo Geoslam is very fast and made very easy the enormous fidelity of a 3D massive cloud point adquisition, but finally it is a black box on how to solve the final point cloud. For this reason, authors made a first step in the explained technology studying specific pathways (table 1) with not so long distances to reduce bias and incorrect attitudes. Faro Scene helped connecting the 6 different evaluations of the could to cloud described on the point 2.3 Method and also drawn on Figure 3.

Line 247 -> Figure 9(b)

Modified in the reviewed manuscript.

You did not explain much about Figure 9. Please explain more on this. Is it filtering effect, right?

Yes, the filtering effect reduce the incomplete or unusable trees, normally because of they are in the limit of the study area or has a lower than 10cm diameter. In the figure9, side a) shows all trees (378) and side b) only the 258 trees with a DBH higher than 0.1m or cilynder complete. We included this explanation in the footprint of the figure 9: “This can be seen by (b) the greater number of trees with DBH of < 0.1 m, and trees in the zone outside the study area that could not be fitted to a geometric shape as they had very few points in their section. Figure 12, please show the unit of X and Y in centimeter instead of meter”.

Nice work!

Reviewer 2 Report

Laser scanning methods are currently widely used in many fields and I think that the knowledge how to conduct the scanning process, the post-processing of data acquired and the final data analysis, is well recognized at the level of individual trees or tree groups remotely sensed detection. Therefore, I do not find a scientific purpose in this work. In my opinion, the presented manuscript contains more the technical report content and the quality assessment of usefulness of point clouds and algorithms than the scientific focus which is expected by the Forests Journal.

If the authors will focus on the scientific problem, I'm always ready to review their manuscript.

Author Response

Laser scanning methods are currently widely used in many fields and I think that the knowledge how to conduct the scanning process, the post-processing of data acquired and the final data analysis, is well recognized at the level of individual trees or tree groups remotely sensed detection. Therefore, I do not find a scientific purpose in this work. In my opinion, the presented manuscript contains more the technical report content and the quality assessment of usefulness of point clouds and algorithms than the scientific focus which is expected by the Forests Journal.

If the authors will focus on the scientific problem, I'm always ready to review their manuscript.

We are thankful for your appreciate comments that has helped to improve our document.

In this study, we compared the standard and traditional method used for tree inventories with the tree calliper tool with a mobile laser scanner performance that allowed a new quickliest and more efficient methodology for this type of studies. For a complete study, we evaluate the accuracy of the estimates of diameter of trees with three fitting algorithms (RANSAC, Monte Carlo and Optimal Circle). The result of the study is an original research that uses a new methodology and provides an important new quantity of information in its discussion.

In the introduction section, the study has been placed in a broad context, highlighting why it is important. Subsequently, the current state of the research field has been examined and key publications have been cited. There are similar studies in the area of research that have been cited having more than 200 citations, which gives an idea of the importance and relevance of the topic treated.

For all the above reasons, we believe that the study has an appropriate structure and sufficient theoretical, methodological and analytical consistency to be considered as a scientific article. Authors have designed a specific methodology that has been evaluated, compared and discussed with similar scientific studies, obtaining important results that allows to consider our work as a new and optimize tool for the management and conservation of a historic garden.

Once more, thank you very much for your time. We hope we have answered your questions. Should you have any further suggestions, we would be happy to make additional changes to improve the manuscript. Thank you.

Reviewer 3 Report

Dear authors,

I have read your manuscript and think you have carried out a very nice study in the environment of a historical garden. I have some notes for you in pdf attached and would like to kindly ask you to read them and try to improve the manucsript. Sometimes, I had a problem to understand the methods etc, but I think it is due to language. Please, if it is possible, I strongly advise you to perform proffessional English editing, as it can really improve the clarity and comprehension of the text. Maybe, it can be the reason of the irrelevance of some of my notes.

Thank you!

Author Response

We are grateful for your comments that have helped improve our document. We have considered all your suggestions.

Line 34. We have included a reference to the parameters and methods for preserving a cultural landscape.

  • Merlos Romero, M. (2013). Paisaje cultural de Aranjuez: parámetros para un plan de gestión. América Patrimonio (5), 24-39.

Line 38. We have included a reference to studies where there have been difficulties due to occlusion in collecting data by means of laser scanner.

  • Herrero-Tejedor, T. R., Arques Soler, F., Lopez-Cuervo Medina, S., Cabrera, M. R. d. l. O., & Martin Romero, J. L. Documenting a cultural landscape using point-cloud 3d models obtained with geomatic integration techniques. The case of the El Encin atomic garden, Madrid (Spain). Plos One 2020, 15(6). doi:10.1371/journal.pone.0235169

Line 42. We have added a reference to studies on methodologies applied to forest inventories and cultural heritage.

  • Liang, X. L., Hyyppa, J., Kaartinen, H., Lehtomaki, M., Pyorala, J., Pfeifer, N., . . . Wang, Y. S. International benchmarking of terrestrial laser scanning approaches for forest inventories. Isprs Journal of Photogrammetry and Remote Sensing 2018, 144, 137-179. doi:10.1016/j.isprsjprs.2018.06.021

  • Cazzani, A., Zerbi, C. M., & Brumana, R. MANAGEMENT PLANS AND WEB-GIS SOFTWARE APPLICATIONS AS ACTIVE AND DYNAMIC TOOLS TO CONSERVE AND VALORIZE HISTORIC PUBLIC GARDENS. 27th Cipa International Symposium: Documenting the Past for a Better Future 2019, 42-2(W15), 291-298. doi:10.5194/isprs-archives-XLII-2-W15-291-2019

Line 49: Given the editorial line of Forests, we wish to highlight areas of interest to the journal.

Line 52: We have added the reference to the study containing this statement.

  • Yrttimaa, T.; Luoma, V.; Saarinen, N.; Kankare, V.; Junttila, S.; Holopainen, M.; Hyyppa, J.; Vastaranta, M. Structural Changes in Boreal Forests Can Be Quantified Using Terrestrial Laser Scanning. Remote Sensing 2020, 12, doi:10.3390/rs12172672.

Line 55: We have modified and placed the name first and the abbreviation in brackets: structure from motion (SfM)”.

Line 82: As the reviewer correctly points out, the TLS obtains spatial data. In the case of tree inventories, the three-dimensional object is a cylinder that is not homogeneous along its whole height. The diameter (DBH) is calculated in different ways depending on the algorithm used. The RANSAC algorithm uses the cylinder as a geometric shape, and the Monte Carlo and Optimal Circle algorithms randomly seek circles according to the two most distant points until they achieve the required precision.

Line 85: We have included the bases of each of the three algorithms used by the authors:

…three fitting algorithms: RANSAC, which calculates the diameter using the geometric revolution of a circle that defines the best cylinder automatically computed by our algorithm; Montecarlo, which uses vertical differential circles adjusting the TLS point clouds until the required accuracy; and Optimal Circle, which calculates the diameter of the trees by fitting all the possible circles that obviously contain the diameter.

Line 99: Modified in the reviewed manuscript.

Line 104: Figure 1 has been modified.

Lines 109-114: Modified in the reviewed manuscript.

Line 116: Modified in the reviewed manuscript.

Line 177: Figure 3 has been modified

Line 194: Figure 4 has been modified

Line 195: It is true that this is a more general figure, but it represents a transversal section of the historic garden in which measurements can be taken to obtain the proportions of the recorded objects, which is precisely what is obtained by the TLS. Figures 5 and 7 provide greater detail, and include three-dimensional representations of the point cloud.

Line 209: The initial cloud had 28 084 808 points (see Table 1).

Line 212: “Level” has been changed to “height difference”.

Line 230: We have improved the description in Figure 7, incorporating the intervals ([0-0.25], [0.25-0.50], [0.50-1.00], [1.00-…])

Lines 243-250: Modified in the reviewed manuscript.

Line 258: Thank you for this comment. We have included a technical description of these three algorithms in the introduction.

Line 282: We have included your suggestion.

Line 289: We have included your recommendation to use “sample” instead of “specimen” for this case, which refers to a generic tree.

Lines 349-351: The great advantage of this methodology is the rapid acquisition of data and the capacity to automate the identification of the trees and their parameters for inventory with a significant reduction in errors.

Line 361: Modified in the reviewed manuscript.

Line 376: Modified in the reviewed manuscript.

Round 2

Reviewer 2 Report

The presented manuscript brings the same version like the first. So, my opinion does not change. The manuscript requires scientific purpose and English proofreading.

Author Response

We are very grateful for your comments, which have helped us improve our document. We still believe we have not sufficiently explained our comments on the scientific method used to develop our research:

We have investigated different methods for using a mobile TLS working with a new Slam measurement procedure that improves the usual method used by TLS (Static Laser).

Current advances in Slam are an opportunity to use this procedure and equipment with automatic methods to improve the usual TLS (which is static and slow procedure) and manual measurements (with a calliper, which is precise but slow) with new automatic procedures. We have included three different algorithms to study and research a new workflow for these proposes.

We looked at the state of the research with an introduction section, and found several works with over 200 citations, which gives an idea of the importance and relevance of this topic.

We believe this study has a suitable structure and the sufficient theoretical, methodological and analytical consistency to be considered as a scientific article. We have designed a specific methodology that has been evaluated, compared with similar scientific studies and discussed. Similar findings, some better or worse than our own, can be verified in parallel studies that have, like ours, obtained important results. Our work is a new and optimised tool for the management and conservation of a historic garden but follows the same line as other studies.

For this reason we would be grateful if you would evaluate our work, as we believe it could be improved by the incorporation of your comments, which are sure to be very interesting. Thank you in advance for your time.

Reviewer 3 Report

Dear authors,

your manuscript is much better now. Please correct two items:

Figure 9: meteres -> m in the scalebar

Latin names are not correct in the sense of fonts, name of the plant in italics and the name abbreviation fo the man who named the plant not in italics.  I attached a picture for you to understand better.

Thank you and goog luck with your article!

Author Response

We are grateful for your comments that have helped improve our document. We have considered all your suggestions.

  • Line 249: We have improved the sentence following your suggestions: “Figure 9: meters -> m in the scalebar”.
  • Line 100. Modified in the reviewed manuscript according to the web site: https://www.ipni.org/
  • Line 109. Modified in the reviewed manuscript according to the web site: https://www.ipni.org/

Once more, thank you very much for your time. We hope we have answered your questions. Should you have any further suggestions, we would be happy to make additional changes to improve the manuscript. Thank you.